# Co-transcriptional splicing efficiency is a gene-specific feature that can be regulated by TGFβ

Elena Sánchez-Escabias[1], José A. Guerrero-Martínez[1✉] & José C. Reyes [1✉]

Differential splicing efficiency of specific introns is a mechanism that dramatically increases protein diversity, based on selection of alternative exons for the final mature mRNA. However, it is unclear whether splicing efficiency of introns within the same gene is coordinated and eventually regulated as a mechanism to control mature mRNA levels. Based on nascent chromatin-associated RNA-sequencing data, we now find that co-transcriptional splicing (CTS) efficiency tends to be similar between the different introns of a gene. We establish that two well-differentiated strategies for CTS efficiency exist, at the extremes of a gradient: short genes that produce high levels of pre-mRNA undergo inefficient splicing, while long genes with relatively low levels of pre-mRNA have an efficient splicing. Notably, we observe that genes with efficient CTS display a higher level of mature mRNA relative to their pre-mRNA levels. Further, we show that the TGFβ signal transduction pathway regulates the general CTS efficiency, causing changes in mature mRNA levels. Taken together, our data indicate that CTS efficiency is a gene-specific characteristic that can be regulated to control gene expression.

---

[1] Centro Andaluz de Biología Molecular y Medicina Regenerativa-CABIMER, Consejo Superior de Investigaciones Científicas-Universidad de Sevilla-Universidad Pablo de Olavide (CSIC-USE-UPO), Avenida Americo Vespucio, 41092 Seville, Spain. ✉email: jose.guerrero@cabimer.es; jose.reyes@cabimer.es

In eukaryotic cells, steady-state levels of mRNAs are determined by the regulated rates of a number of processes, including transcription, mRNA maturation, and mRNA degradation[1]. One of the most important steps of mRNA maturation is splicing, during which introns from precursor messenger RNAs (pre-mRNAs) are removed, and exons are joined together, to produce spliced mRNAs[2,3]. Splicing has been extensively investigated as a mechanism of increasing gene-product diversity by using alternative splice sites to include or exclude particular alternative exons[4–6]. However, much less is known about the consequences of splicing efficiency in the final gene expression levels.

The splicing efficiencies of specific introns are influenced by the 5′- and 3′-splicing sites sequences, specific sequences such as the intronic branch-point sequence, intronic and exonic splicing enhancers[2,3,7], nascent RNA folding[8,9] and other mRNA processing reactions (including 5′-end capping or 3′-cleavage and polyadenylation) (reviewed in ref. [10]). Researchers have proven (using different strategies) that nascent RNA is mostly spliced during transcription elongation (co-transcriptional splicing, CTS)[10–19]. The kinetics of CTS is currently a debated issue, with different groups reporting apparently contradictory results based mostly on single-molecule sequencing of nascent RNA. For instance, the Neugebauer laboratory showed that splicing occurs when RNA polymerase II (RNAPII) has transcribed between 26 and 300 nucleotides downstream of the 3′ splicing site (ss), often during transcription of the downstream exon both in yeast[20] and humans[21]; in contrast, the Churchman group reported that human introns were spliced much later, when RNAPII has transcribed about 4 kb downstream of the 3′-ss[22]. Recently, Sousa-Luis et al.[23] have found both behaviors: immediate and delayed CTS. However, what determines whether one or the other mechanism is used is unknown.

The co-transcriptional nature of splicing favors the effect of RNAPII elongation rate on alternative splicing (reviewed in ref. [5]). In yeast, slow RNAPII elongation increases both CTS and splicing efficiency[24,25]. In contrast, faster RNAPII elongation causes strong defects in splicing, suggesting that splicing can become rate limiting when transcription is fast[20,25]. In mammalian cells, an altered elongation speed causes enhanced inclusion or skipping of specific alternative exons, as well as intron retention[26–28], but no changes in whole transcript expression levels have been reported. In fact, to what extent CTS efficiency is an intron- and/or a gene-specific characteristic is not clear. Furthermore, is gene CTS efficiency a parameter that can be regulated by developmental cues and signal transduction pathways or, alternatively, does it mostly depend on gene structural features that cannot be controlled by external factors?

We have addressed these questions by calculating a gene splicing index (GSI) from nascent RNA-enriched RNA-sequencing data of epithelial cells grown in the absence or the presence of the growth factor TGFβ, which promotes a strong change in the transcriptome[29–31]. We studied the relationship between changes in the GSI and the steady-state levels of mature mRNAs, determined by using normal RNA-seq data from the same conditions. Our data suggest that CTS efficiency is a gene-specific characteristic with two extreme behaviors: long, modestly expressed genes are efficiently spliced (high-GSI), while short, highly expressed genes are relatively inefficiently spliced (low-GSI). Furthermore, we show that TGFβ promoted changes in the levels of mature mRNAs associated to changes in GSI.

## Results

### Introns of the same transcript show coordinated co-transcriptional splicing efficiency. To investigate whether CTS

efficiency is coordinated among the different introns of a gene, first we calculated an intron splicing index (ISI) for all introns of expressed genes in the non-transformed mouse mammary epithelial cell line NMuMG[32]. For that, we used published chromatin-associated RNA-seq (ChrRNA-seq) data from our lab[31] (see details in Methods). Chromatin-associated RNA is mostly constituted by nascent RNA associated to elongating RNAPII, as well as full length transcripts not fully spliced[12,33,34]. ISI was calculated in two ways: junction-based ISI (ISI$_j$) and coverage-based ISI (ISI$_c$). ISI$_j$ was determined using reads mapping across the exon boundaries into the adjacent intron sequences (indicative of unspliced introns), and reads mapping across exon–exon junctions (indicative of spliced introns); ISI$_c$ was based in the number of reads mapped to an intron and the reads mapped to the adjacent exons adjusting to intron or exon length (see Fig. 1a and Methods). A high ISI value for an intron is indicative of an efficient CTS. Fig. 1b shows good correlation (Pearson coefficient = 0.5) between ISI$_j$ and ISI$_c$ for 79975 introns among 7523 expressed genes (see also list of ISI$_j$ and ISI$_c$ for all analyzed introns, (Supplementary Data 1). As previously shown in other studies[18,19,21], ISI values decreased with decreasing distance to the polyA sites, in agreement with the co-transcriptional nature of splicing (Fig. 1c and Supplementary Fig. 1a). Also in agreement to other studies, we found that first and last introns tended to be inefficiently spliced (Supplementary Fig. 1b, c). This inefficiency has been related to (i) the high frequency of very large first introns in mammals, and (ii) the fact that efficient splicing of first and last introns may require interactions with capping and cleavage/polyadenylation machinery, respectively[17,35–38].

We noted that, with the exception of a few specific introns in some genes, most introns of a transcription unit presented similar ISI values. Further, ISI levels of introns within the same gene tended to be more similar to each other than those of introns from different genes (see examples, Fig. 1d). Confirming this fact, the variances of both ISI$_c$ and ISI$_j$ values of introns within the same gene were smaller than those from randomly selected introns (p-value = $4.81 \times 10^{-11}$ and $2.31 \times 10^{-33}$, respectively) (Fig. 1e, f). Similar results were obtained when ISI$_c$ of genes with similar number of introns were used to compute variances (Supplementary Fig. 2). This results suggested that CTS efficiency is, at least in part, a gene-specific trait.

### Gene Splicing Index correlates positively with gene length and negatively with pre-mRNA levels. To gain a better under-

standing of CTS efficiency at the gene level, we computed a new parameter that we called the Gene Splicing Index (GSI). The GSI of a gene was calculated as the log$_2$ ratio between exonic and total pre-mRNA reads (using the longest transcript per gene), and relativized to exons or gene length, respectively (Fig. 2a). Therefore, an elevated GSI indicates an efficient CTS, while a low GSI indicates poor CTS efficiency. The GSI values ranged from −0.29 to 6.23, with an average of 1.35 (see list of GSI values for all expressed genes, Supplementary Data 2). GSI values were then sorted in increasing order and binned into deciles (Fig. 2b). Strikingly, the first decile GSI values differed very significantly from last decile GSI values (p-value < $10^{-300}$), and the corresponding genes were considered low- and high-GSI genes, respectively (see examples of high- and low-GSI genes, Fig. 2c). Determination by quantitative reverse transcription PCR (RT-qPCR) of specific intronic and exonic sequences in the pre-mRNA of one low-GSI gene (Id3) and two high-GSI genes (Inadl and Utrn) using chromatin-associated RNA, confirmed a high or low proportion of unspliced introns in the low or high-GSI genes, respectively (Fig. 2d). Notably, functional analysis using Gene Ontology (GO) revealed that low-GSI genes were strongly

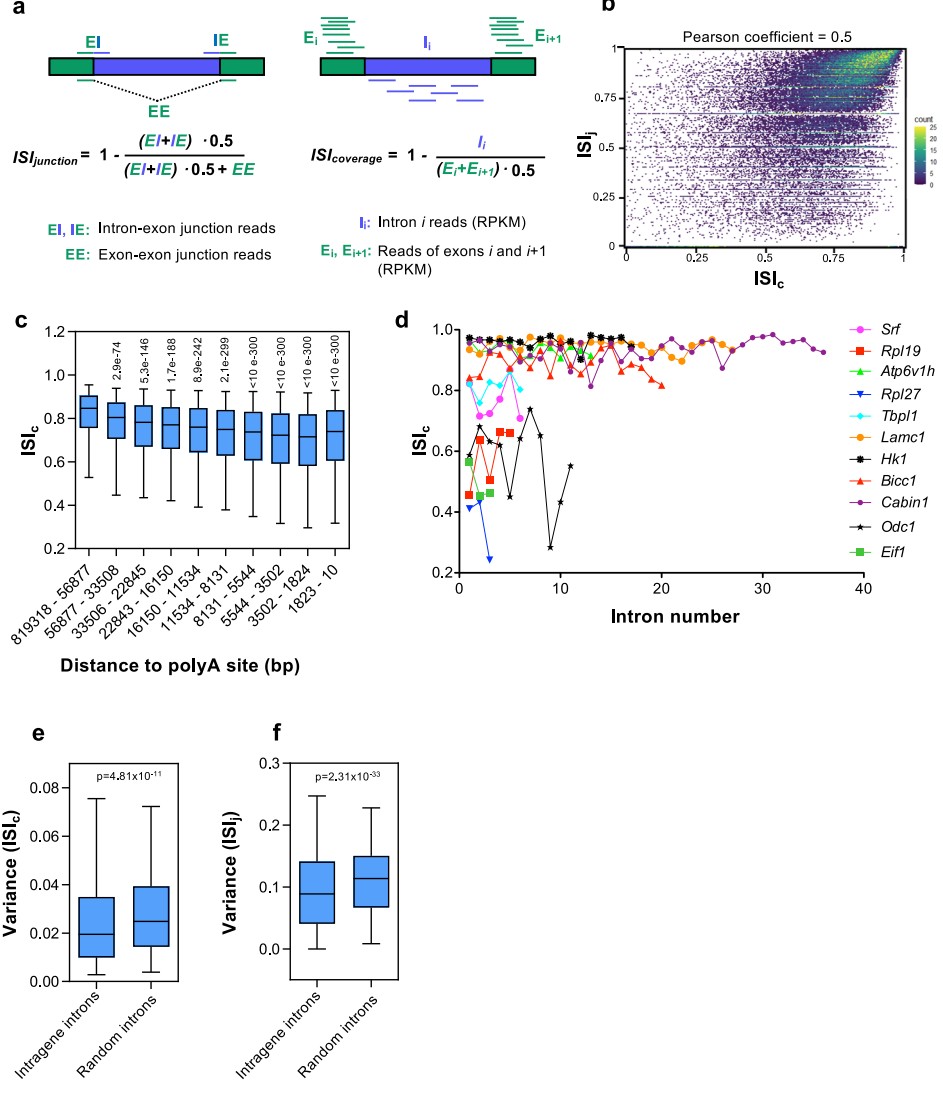

**Fig. 1 Intron splicing efficiency. a** Schematics outlining the two approaches used to measure the intron splicing index (ISI): junction-based ISI ($ISI_j$) and coverage-based ISI ($ISI_c$). RPKM, reads per kilobase per million mapped reads. **b** Scatter plot of $ISI_j$ and $ISI_c$ values correlation. Each point represents the $ISI_j$ and $ISI_c$ of a certain intron. **c** Effect of intron to polyA site distance on $ISI_c$. Introns were divided into ten deciles depending on their distance to the polyA site. Unpaired Student's t-test p-values of the indicated decile with respect to the first decile are shown. **d** $ISI_c$ level of the indicated introns of eleven different genes. **e**, **f** Variance of $ISI_c$ (**e**) and $ISI_j$ (**f**) values across introns within the same gene or the same number of randomly sampled introns. Unpaired Student's t-test p-values are shown. Sample size (n) of all sets of data are provided in Supplementary Data 4.

enriched in transcription and translation GO categories, whereas high-GSI genes were enriched in transport, metabolism and cell adhesion categories (Fig. 2e).

Next, we investigated the relationship between GSI and total nascent pre-mRNA levels. Interestingly, GSI negatively correlated with total pre-mRNA levels, (Pearson coefficient = −0.17; $p \leq 0.0001$) with a very significant difference between low-GSI and high-GSI genes ($p = 2.81 \times 10^{-31}$) (Fig. 3a, b), suggesting that highly transcribed genes tend to have a deficient CTS efficiency, which might be associated with the difficulties of the splicing machinery as a limiting factor[39] for coping with a high rate of transcription.

We then explored the effect of gene length on the GSI. A positive correlation was observed between gene length and GSI value, with a very significant difference in gene length between low-GSI and high-GSI genes ($p = 1.72 \times 10^{-88}$) (Fig. 3c, d). Similar positive correlation was also observed between the GSI and intron number ($p = 8.37 \times 10^{-133}$) (Fig. 3e, f). Since long genes tend to contain more introns, we tried to dissect the effect

of both parameters on the GSI, by computing (i) the effects of gene length in genes with a similar number of introns, and (ii) the effects of introns number in genes of a similar size. We found that gene length was positively correlated to GSI even among genes with similar number of introns (Pearson coefficients 0.3–0.5) (Supplementary Fig. 3). Similarly, the number of introns was also positively correlated to GSI for genes with similar size (Pearson coefficients 0.2–0.4) (Supplementary Fig. 4). These data indicated that gene length, intron number and pre-mRNA levels are important factors for gene CTS efficiency.

We then investigated how the combination of gene length, and gene expression on the same genes influence the GSI parameter. For that, we divided the gene population into ten bins according to their length in increasing order ($L_i$, with $i = 1, 2, …, 10$) and into another ten bins according to their pre-mRNA levels in increasing order ($E_j$, with $j = 1, 2, …, 10$). A matrix ($L \times E$) was then constructed by assigning genes to the corresponding positions $a_{i,j}$ according to their respective length ($L_i$) and pre-mRNA level ($E_j$). Supplementary Fig. 5 shows a heatmap with the

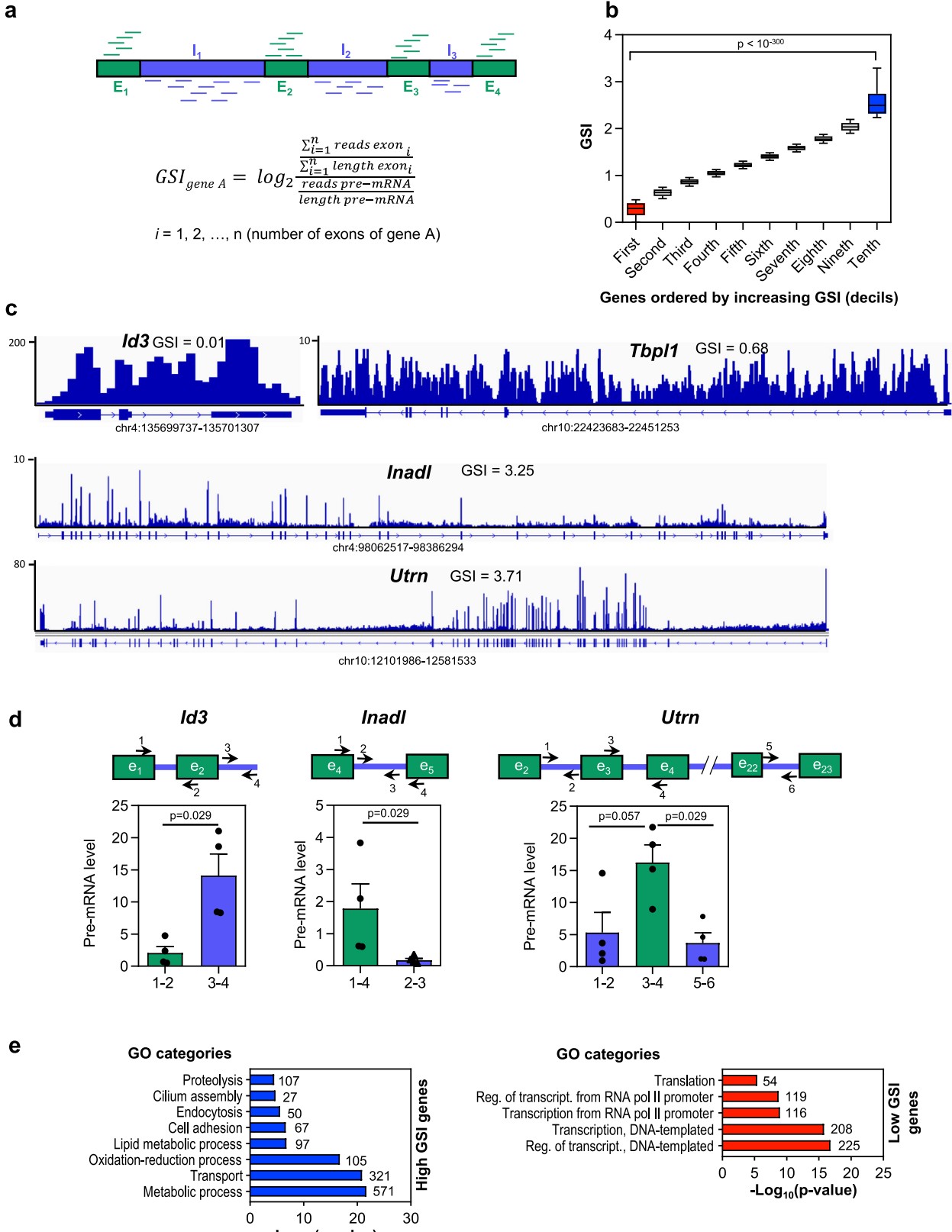

**Fig. 2 Gene splicing efficiency. a** Schematics outlining the approach used to measure the Gene Splicing Index (GSI). RPKM, reads per kilobase per million mapped reads. **b** GSI values ordered from low to high GSI and binned into ten deciles. Low-GSI genes (i.e., genes in the first decile) are indicated in red, and high-GSI genes (i.e., genes in the last decile), in blue. Student's *t*-test *p*-values of the indicated comparison are shown. **c** Chromatin-associated RNA-seq (ChrRNA-seq) IGV snapshots are shown for genes with a low GSI (*Id3* and *Tbpl1*) or a high GSI (*Inadl* and *Utrn*). **d** RT-qPCR determination of intronic and exonic levels of one low-GSI gene (*Id3*) and two high-GSI genes (*Inadl* and *Utrn*) using chromatin-associated RNA. Represented values are mean ± SEM of four (*n* = 4) independent biological replicates. Unpaired two-tailed Mann–Whitney *p*-values of the indicated comparison are provided. **e** Functional analysis using Gene Ontology (GO) of high-GSI genes and low-GSI genes. Number of genes in each category is shown. Sample size (*n*) of all sets of data are provided in Supplementary Data 4.

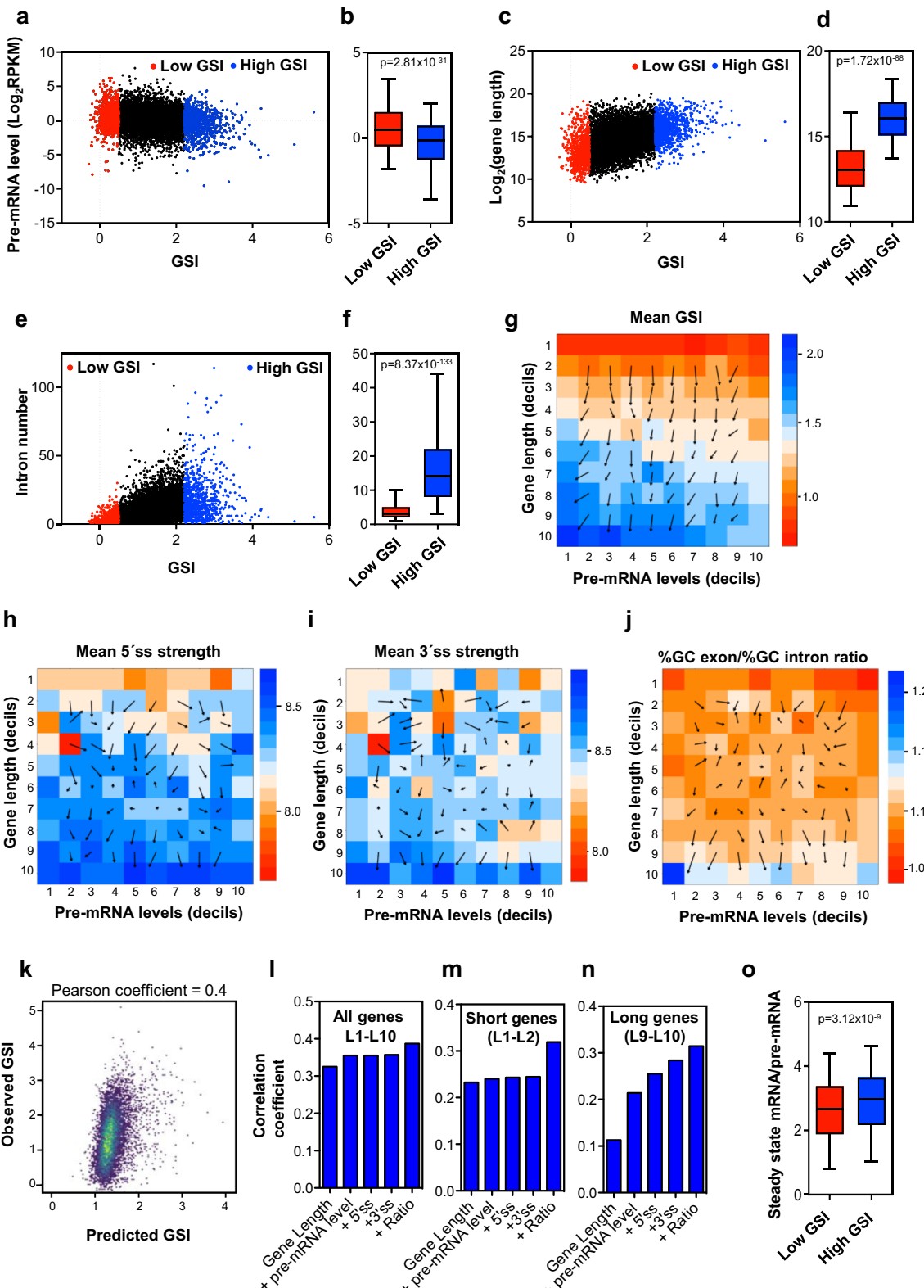

number of elements (genes) for each position $a_{i,j}$. The mean GSI values of the genes corresponding to each position of the matrix were computed and represented in a heatmap (Fig. 3g). As expected from our previous analysis, the mean GSI of $a_{10,1}$, corresponding to the largest genes ($L_{10}$), and the lowest expression ($E_1$) displayed the highest mean GSI value. Furthermore, a gradient vector field analysis of the $L \times E$ matrix indicated

the direction in which the matrix varies more quickly and the rate of variation in that direction. These analyses revealed that gene length is a stronger determinant of GSI value than pre-mRNA levels throughout all lengths and pre-mRNA level ranges.

Next we considered how other structural variables, which theoretically affect splicing, distribute along this matrix. The 5′ and 3′ splice site (ss) motifs are obvious players that affect

**Fig. 3 Characterization of gene CTS efficiency with respect to gene structural variables and pre-mRNA levels. a** Correlation between nascent pre-mRNA levels and GSI values. Data from low-GSI or high-GSI genes are depicted in red or blue, respectively. **b** Boxplot of pre-mRNA levels in low-GSI and high-GSI genes. **c** Scatter plot of gene lengths versus GSI values. **d** Boxplot of gene lengths in low-GSI and high-GSI genes. **e** Scatter plot of intron numbers versus GSI values. **f** Boxplot of the intron numbers in low-GSI and high-GSI genes. **g–j** Heatmap matrices of mean GSI levels (**g**), 5′ ss motif strength (**h**), 3′ ss motif strength (**i**) and exon/intron GC content ratio (**j**), depending on gene length ($L$) and pre-mRNA level ($E$). The $L \times E$ was constructed by assigning genes to the corresponding positions $a_{i,j}$ according to their respective length decil ($L_i$) and pre-mRNA level decil ($E_j$). The number of genes in each position of the matrix is shown in Supplementary Fig. 5. A gradient vector field that indicates the direction in which the matrix varies more quickly, and the rate of variation in that direction, is shown. **k** Correlation between observed GSI and predicted GSI values, determined using a multiple linear regression model with all considered variables. **l–n** Pearson correlations coefficients of the linear regression model for GSI estimations using increasing subsets of the indicated variables for all expressed genes (**l**), short expressed genes (deciles L1-L2 of matrices panels **g–j**) (**m**) or long expressed genes (deciles L9-L10 of matrices panel **g–j**) (**n**). **o** Boxplot of (mature mRNAs level)/(pre-mRNA level) ratios in low-GSI and high-GSI genes. **b**, **d**, **f**, **o** Unpaired Student's $t$-test $p$-values are shown. Sample size ($n$) of all sets of data are provided in Supplementary Data 4.

splicing efficiency at the intron level[2,3]. The intronic 5′ and 3′ ss strengths were computed using a maximum entropy model[40]. Interestingly, the gene average of the 5′ ss strength was low in small genes ($L_1$ and $L_2$) and tended to be higher in long genes ($L_9$ and $L_{10}$), with no clear tendency in the intermediate lengths. No effect of pre-mRNA levels was observed (Fig. 3h). A high 3′ ss strength was only observed in very long genes ($L_{10}$) (Fig. 3i). Differential exon–intron GC content is known to influence exon inclusion, especially in long genes[41]. Therefore, we also studied the distribution of this parameter in the $L \times E$ matrix. Our data show that gene exon/intron %GC ratio was higher than the average only in long genes ($L_9$ and $L_{10}$), and was lower than the average in very short genes ($L_1$), reaching a maximum in very long genes with very low pre-mRNA levels, coinciding with the most efficient CTS genes (Fig. 3j).

Next we performed a multiple linear regression model to estimate the GSI including all the parameters described above. While a good correlation coefficient was obtained between predicted and observed GSI values (Pearson coefficient = 0.4), these variables only accounted for 14.97% of the variance of the GSI (Fig. 3k). Progressive inclusion of the different variables in the model again demonstrated that gene length is, by far, the strongest contributor to the model (Fig. 3l). Matrices shown in Fig. 3h, i, and j suggest that 5′ and 3′ ss strength and the ratios of exon/intron GC content change when comparing long or short genes with respect to intermediate lengths. Indeed, when our linear regression model was tested only with short ($L_1 + L_2$) or long genes ($L_9 + L_{10}$), the contribution of these variables to the model was much more important, especially for long genes (Fig. 3m, n).

Taken together, our data indicate that CTS efficiency is a continuum variable, strongly dependent on gene length and less strongly dependent on pre-mRNA level, with two extreme behaviors: highly transcribed short genes encoding mostly transcription factors or translation proteins, which display inefficient splicing, and long genes mostly encoding metabolic enzymes or transport proteins with a not very high level of pre-mRNA. Interestingly, this last set of genes display strong 5′ ss and 3′ ss motifs and high ratios of exon/intron GC content, suggesting that these features have been subjected to a strong selective pressure in long genes to be efficiently spliced.

Finally, we investigate the consequence of the GSI on the level of mature mRNAs, with respect to the nascent pre-mRNA level. The steady-state of the mature mRNA levels were determined by using standard RNA-seq data under the same conditions[31]. Importantly, we observed that the ratio of mature mRNA level versus the pre-mRNA level was significantly higher in genes with high-GSI values than in genes with low-GSI values (Fig. 3o), indicating that CTS efficiency influences final expression levels.

**TGFβ treatment causes transitory and permanent GSI changes**. Our linear regression model only explains around 15% of the GSI

variance, indicating that other unknown factors should contribute to GSI determination and eventually to its regulation. Therefore, we wondered whether the GSI can be regulated by a signal transduction pathway. To analyze that, we used ChrRNA-seq data from cells treated with the growth factor TGFβ for 2 h or 12 h, which provokes dramatic changes in the transcriptome that induce the epithelial-to-mesenchymal transition in NMuMG cells[29–31]. The GSI values for the two different time points after TGFβ addition, from two independent biological replicates, were determined, and differential GSI values (ΔGSI) at 2 h or 12 h versus vehicle-treated (control) cells were computed using LIMMA differential analysis[42] (Supplementary Data 3). Overall, 125 or 82 genes showed an increased (ΔGSI ≥ 0.5; $p \leq 0.05$) or decreased (ΔGSI ≤ –0.5; $p \leq 0.05$) splicing efficiency, respectively, at the 2 h time point, while 53 and 65 genes showed an increased or a decreased splicing efficiency, respectively, at the 12 h time point. Notably, at the 2 h time point, most genes that presented a decrease of splicing efficiency were induced by TGFβ. In contrast, most of the genes that increased CTS efficiency were repressed by TGFβ (Fig. 4a, b). This was in agreement with our previous results, which indicated that nascent pre-mRNA levels are negatively correlated with the GSI. We also observed that a decreased or increased CTS efficiency at 2 h was transient. In other words, most of the genes that had increased or decreased GSI values at the 2 h time point (as compared to the TGFβ untreated cells) returned to their baseline levels at the 12 h time point (Fig. 4c, d). These data suggest that an increase or decrease of gene transcription caused by TGFβ addition promotes a transient deficiency or improvement of the CTS efficiency, respectively; however, later, the splicing machinery adapts to the new transcription rate causing the recovery of the original GSI values. Fig. 4e shows an example of this behavior in the gene *Ncam1*. ChrRNA-seq data showed that *Ncam1* nascent pre-mRNA increased 2 h after TGFβ addition; notably, the exon signal did not change, leading to a decrease of the GSI (from 2.21 to 1.11) at this time point. However, the increased exon signal at the 12 h time point caused the recovery of the GSI value. This slow splicing kinetics has consequences for the mature mRNA levels (determined by RNA-seq), which only increased at the later time point (Fig. 4e). A similar behavior was observed for the TGFβ-induced gene *Cacna2d1* (Supplementary Fig. 6a), and an inverse behavior, for the TGFβ-repressed gene *Angpt1* (Supplementary Fig. 6b). Consistent with this delayed splicing efficiency adaptation, most of the genes that have a decreased GSI value at 2 h after TGFβ addition presented higher levels of mature mRNA at 12 h than at 2 h (after TGFβ addition) (Fig. 4f). Inversely, most of the genes that gained in the GSI value at 2 h after TGFβ addition presented lower levels of mature mRNA at 12 h than at 2 h (after TGFβ addition) (Fig. 4g). In summary, these data suggest that the GSI changes after 2 h of TGFβ are transitory and are a consequence of the slow adaptation to the new transcription rate of the regulated genes.

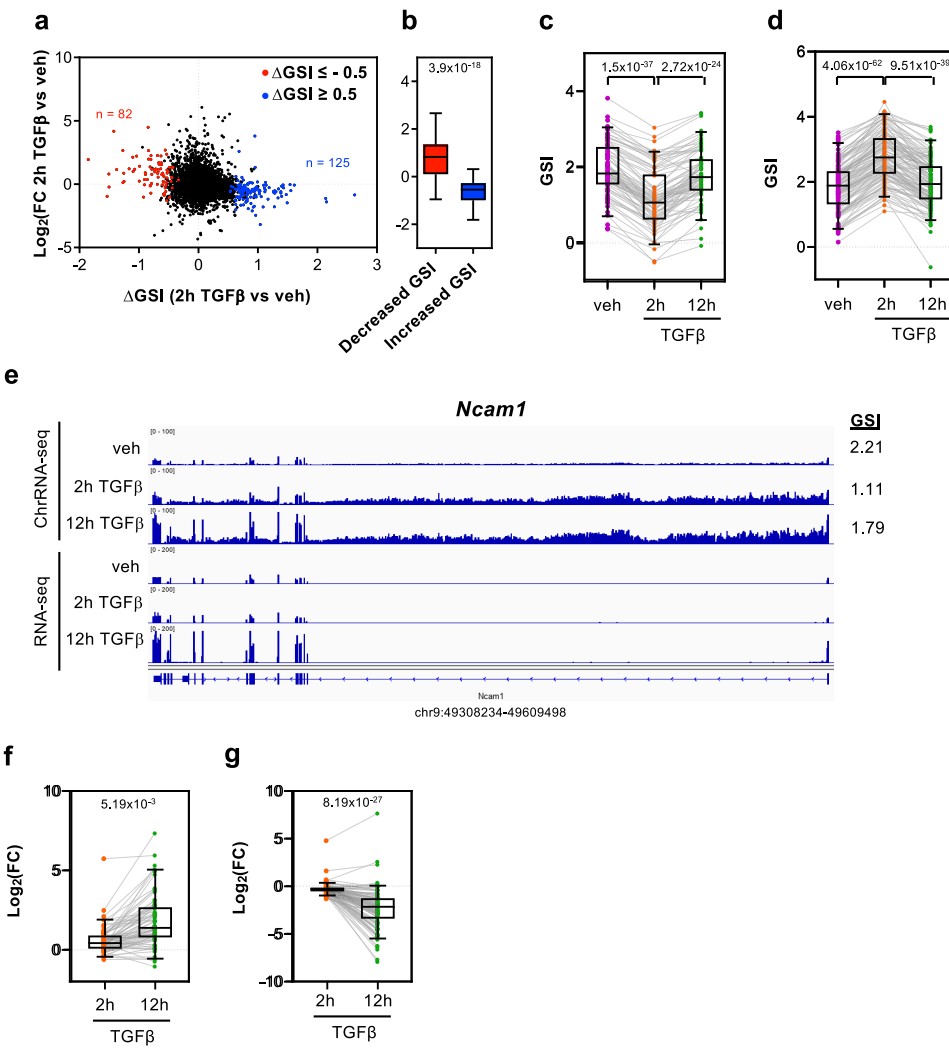

**Fig. 4 Change of GSI (ΔGSI) after 2 h of TGFβ treatment. a** Scatter plot of nascent pre-mRNA level changes (log$_2$FC) versus GSI changes (ΔGSI) after 2 h of TGFβ treatment. Genes with decreased GSI (ΔGSI < –0.5, $p < 0.05$) or increased GSI (ΔGSI > 0.5, $p < 0.05$) are depicted in red or blue, respectively. **b** Boxplot of nascent pre-mRNA level changes (log$_2$FC) after 2 h of TGFβ treatment versus vehicle, for genes with decreased or increased GSI values. Unpaired Student's $t$-test $p$-values are shown. **c, d** GSI levels at the three conditions tested (vehicle, 2 and 12 h of TGFβ treatment) for genes with decreased GSI (**c**) or increased GSI (**d**) after 2 h of TGFβ treatment. **e** ChrRNA-seq and RNA-seq IGV snapshot of *Ncam1* gene as example of a transient change of GSI after 2 h of TGFβ. **f, g** Changes of mature mRNAs level (log$_2$FC) after 2 h or 12 h of TGFβ of genes with decreased (**f**) or increased GSI (**g**) at the 2 h TGFβ time point. **c, d, f, g** Paired Student's $t$-test $p$-values of the comparison between the indicated distributions are shown. Sample size ($n$) of all set of data are provided in Supplementary Data 4.

Next, we analyzed genes that changed their GSI at 12 h after TGFβ addition. At this time point, master regulators of the epithelial to mesenchymal transition process are expressed, and morphological changes are already visible[31,32]; therefore, we considered that these are stable changes of the GSI. In this case, and in contrast to what happened at the 2 h time point, genes with increased GSI values were mostly upregulated by TGFβ, while genes with decreased GSI values were mostly down-regulated by TGFβ (Fig. 5a, b). This behavior challenges the general tendency described above (see Fig. 3a, b, and g) about an inverse correlation between GSI and pre-mRNA levels. GSI values at 2 h of these genes were unchanged or slightly changed, in the same sense, at the later time point (Fig. 5c, d). We next analyzed in detail two genes representative of this behavior, *Wdr1* and *Csrp1*. For that, the levels of pre-mRNA in nascent chromatin-associated RNA, and levels of mature mRNA in the cytoplasmic fraction of the same samples, were determined by RT-qPCR, using exon–exon or exon–intron amplicons, at 2 h or 12 h after

addition of TGFβ or (as a control) vehicle. Nascent RNA-seq data indicated that the increase of the GSI in *Wdr1* and *Csrp1* genes was caused by both a moderated increase of exonic reads and a decrease of intronic reads (Fig. 6a, e). In agreement, RT-qPCR experiments demonstrated an increased exonic signal and a decreased intronic signal at 12 h after TGFβ addition with respect to vehicle addition (Fig. 6b, f). Consistently, the exon/intron ratio increased dramatically (6.45-fold for *Wdr1* and 12.30-fold for *Csrp1*) at the 12 h time point (Fig. 6c, g), coinciding also with a strong increase (4.70-fold for *Wdr1*, and 56.51-fold for *Csrp1*) in the level of both mature mRNAs in the cytoplasm (Fig. 6d, h). These data indicate that general CTS efficiency of these two genes is regulated by TGFβ.

## Discussion

Splicing is an essential step of gene expression in eukaryotes that dramatically increases protein diversity by selecting alternative

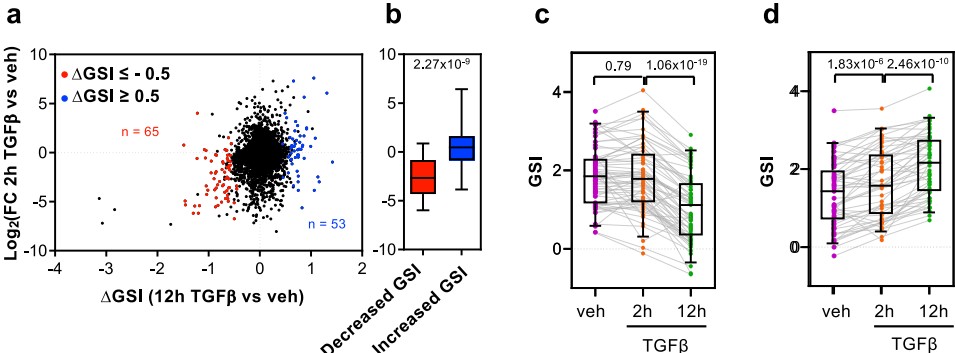

**Fig. 5 Effect of 12 h of TGFβ treatment on GSI changes (ΔGSI). a** Scatter plot of nascent pre-mRNA level changes ($log_2FC$) versus GSI changes (ΔGSI) after 12 h of TGFβ. Genes with decreased GSI (ΔGSI < −0.5, $p < 0.05$) or increased GSI (ΔGSI > 0.5, $p < 0.05$) are depicted in red or blue, respectively. **b** Boxplot of nascent pre-mRNA level changes ($log_2FC$) after 12 h of treatment with TGFβ versus vehicle in genes with decreased GSI or increased GSI. Unpaired Student's $t$-test $p$-values are shown. **c, d** GSI levels at the three conditions tested (vehicle or 2 and 12 h of TGFβ-treated at the 2 h or 12 h time point) for genes with a decreased GSI (**c**) or an increased GSI (**d**) after 12 h of TGFβ. Paired Student's $t$-test $p$-values of the comparison between the indicated distributions are shown. Sample size ($n$) of all set of data are provided in Supplementary Data 4.

exons for the final mature mRNA. However, whether regulated changes in the splicing efficiency controls the steady-state level of mature mRNA is unclear. From a kinetic point of view, splicing has been considered a limiting step in gene expression, with time estimations for intron removal ranging from 30 s to 1 h[15,43–45]. Specific intronic CTS efficiency is dependent on the distance to the polyA site, consistent with the "first-come, first-served" model of splicing that proposes that the first introns transcribed are the first to be committed for splicing[46] (note that the universal validity of this model is currently under debate[21–23]). Other factors can also affect CTS efficiency at the intron level in animals and plants, such as intron and exon length, intron position, and 5´ss and 3´ss strength[17–19,45,47,48]. Our analysis using nascent pre-mRNA data and two different intron CTS quantification methods ($ISI_c$ and $ISI_j$), provided similar results. We found a remarkable similarity between the CTS of different introns within the same gene. In fact, a coordinated splicing efficiency within a gene has been noted in two recent publications[21,45] using different strategies, but they did not further characterize it, neither studied the potential regulatory implications of this phenomenon. To address this, we defined a gene splicing index (GSI) that reveals the average gene CTS efficiency. We found that, in general, long genes with many introns and with a moderate or low level of nascent pre-mRNA had a high GSI; in contrast, short genes with few introns and a high level of nascent pre-mRNA had a low GSI. Positive correlation between gene length and efficiency is consistent with the co-transcriptional nature of splicing: short genes have a limited time to carry out the CTS process. Given that the average transcription rate in mammals is about 1.5 kb/min[49] and the average splicing half-life estimated from metabolic labeling data is 7–14 min[43,50], introns that have their 3´ ss closer than 21 kb from the transcription termination site may have difficulties to complete splicing before transcription termination. Thus, it is to be expected that genes shorter than about 20 kb have a low GSI. However, we observed that GSI linearly depends on gene length throughout all length ranges. It is possible that longer times are required to complete whole gene splicing. In this sense, Bhatt et al.[34] reported the existence of full length incompletely spliced transcripts in the chromatin-associated pre-mRNA fraction, and similar results have been recently reported using single-molecule sequencing of nascent RNA[22,23].

The effect of nascent pre-mRNA level on CTS efficiency is, however, a more debated issue. In principle, as any other enzymatic process, CTS should follow a saturation kinetics

characterized by a decline in efficiency at high substrate (pre-mRNA) concentration. Furthermore, work from yeast has demonstrated a strong competition between pre-mRNAs for a limited amount of splicing machinery[39]. Despite these facts, Ding and Elowitz[51], using a single-cell imaging system, have reported an "economy of scale" behavior, in which splicing efficiency increases with transcription rate in two specific genes. Several works have reported a small but significant positive correlation between mature mRNA level (from RNA-seq data) and intron CTS efficiency, both in animals and plants[19,45,47,48]. However, whether a high level of mature mRNA is a cause or a consequence of the high CTS efficiency has not been clarified. Given the fact that we observe a negative correlation between the GSI and pre-mRNA levels, but a higher mature mRNA/pre-mRNA ratio in high-GSI versus low-GSI genes, we conclude that a high level of transcription impairs gene CTS efficiency probably by saturation of the splicing machinery, which has a negative effect in gene expression. In contrast, a low level of transcription promotes a more efficient splicing and a positive effect for gene expression. In agreement with our data, Tilgner et al.[19] reported a positive correlation between intron CTS efficiency and mature mRNA, but a negative correlation with the level of RNA polymerase II. Interestingly, we observed that high-GSI genes were strongly enriched in metabolic enzymes, transport, and adhesion proteins while low-GSI genes were enriched in transcription factors and translation proteins. These functional associations are probably due, at least in part, to the different lengths of the genes of each GO functional category. Thus, ribosomal proteins and many transcription factors (especially immediate early genes) are encoded by short genes with a short number of introns[52–54]. In contrast, extracellular matrix, adhesion molecules, endocytosis, and transport categories are often encoded by long genes (see Supplementary Fig. 7). These results seem to define two well-differentiated strategies for gene expression at the extremes of a gradient: one for short, highly transcribed genes, for which pre-mRNA levels are so high that a relatively inefficient splicing has little consequences; and another for long genes with relatively low level of transcription, for which it is important to have a very efficient splicing to maintain acceptable levels of gene expression. This is in agreement with the fact that very long genes tend to have stronger 5′ ss and 3′ ss motifs than short genes and a high exon/intron GC content ratio (Fig. 3h, i, and j). An association of long introns with strong 5′ ss and 3′ ss and a high exon/intron GC content ratio has been previously identified[41,55]. We now extend

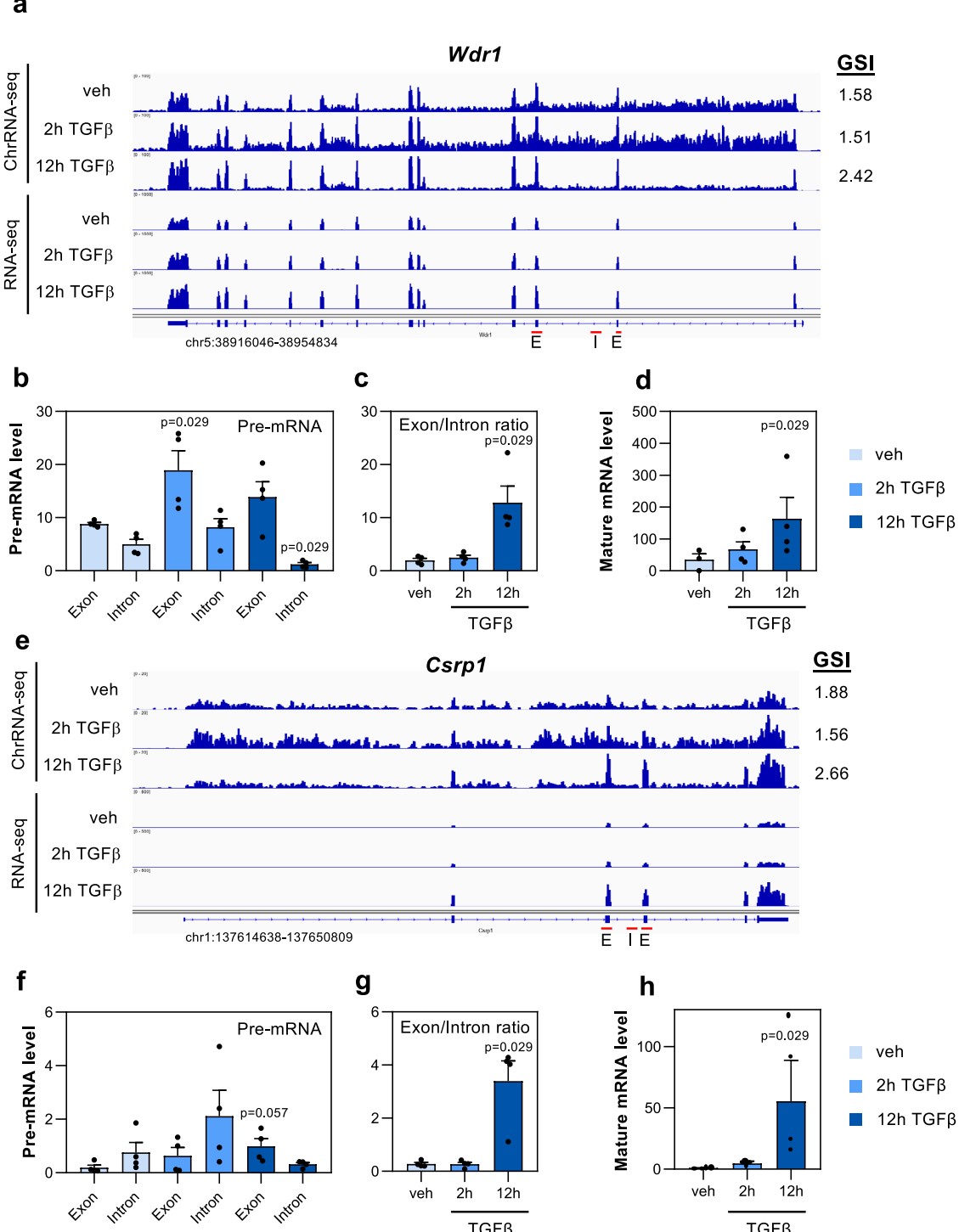

**Fig. 6 Effects on gene expression of GSI changes after 12 h of TGFβ addition. a**, **e** ChrRNA-seq and RNA-seq IGV snapshot of *Wdr1* (**a**) and *Csrp1* (**e**) genes in vehicle or at 2 or 12 h after TGFβ treatment. **b**, **f** Exon and intron levels of *Wdr1* (**b**) and *Csrp1* (**f**) transcripts in nascent chromatin-associated RNA isolated at the three conditions tested: vehicle or at 2 h or 12 h after TGFβ. Levels were determined by RT-qPCR, using exon–exon or exon–intron amplicons (oligonucleotides indicated as red arrows). **c**, **g** Exon/intron ratios for *Wdr1* or *Csrp1* transcripts using data shown in **b**, **f**, respectively. **d**, **h** Mature mRNA levels of *Wdr1* and *Csrp1 genes* at the three conditions tested: vehicle (control) or at 2 h or 12 h after TGFβ addition determined by RT-qPCR using RNA from the cytoplasmic fraction and exon–exon oligonucleotides (indicated as red arrows). **b**–**d**, **f**–**h** Values represent the mean ± SEM of four (*n* = 4) independent biological replicates. Unpaired two-tailed Mann–Whitney *p*-values of the indicated comparison are shown. Sample size (*n*) of all set of data are provided in Supplementary Data 4.

this association genome-wide to long genes with a good splicing efficiency. From a metabolic and energetic perspective, synthesis of long-gene pre-mRNAs requires massive quantities of nucleotides, energy and time. It seems reasonable to suppose that long genes have been subjected to a strong selective pressure to have a very efficient splicing process.

Our data also suggest that GSI changes promoted by signal transduction pathways, such as the TGFβ pathway, can have strong regulatory consequences. For instance, we found a number of genes with a transient change of GSI, caused by a change of the intronic signal but not in the exonic signal 2 h after TGFβ addition, indicating that, in these genes, splicing is temporally decoupled from transcription. However, at the 12 h time point these genes recovered the original splicing efficiency. This slow adaptation to the TGFβ-promoted change of transcription rate correlates, and maybe causes, a delayed change of mature mRNA levels; thus, it may constitute a splicing-dependent temporal regulation of gene expression.

We also observed stables changes of GSI. After TGFβ treatment, *Wdr1* and *Csrp1* genes presented only small changes in nascent pre-mRNA level but major changes in exon/intron ratios and in cytoplasmic mature mRNA levels. Our data suggest that the strong increase in mature mRNA levels is a consequence of a better splicing efficiency. What factors can be responsible for this change in splicing efficiency? One possibility is that TGFβ affects the RNAPII elongation rate of these genes. The kinetic model of coupling between alternative splicing and transcription elongation proposes that the RNAPII elongation rate influences alternative splicing by affecting the pace at which splice sites and regulatory sequences emerge in the nascent pre-mRNA during transcription. Large differences in elongation rate have been reported between genes, mostly caused by the level of transcription, the exon density, or the chromatin configuration of the gene body (nucleosome density, histone variant composition and histone posttranslational modification), which are ultimately determined by the level of transcription[49,56,57]. An alternative possibility is that certain signal-dependent transcription factors are able to recruit splicing factors to the promoters of their target genes, which in turn can modulate CTS efficiency. For example, a role of nuclear receptors transcription factors in alternative splicing has been well documented[58,59]. Finally, it is also possible that transcription factors recruit certain loci to specific nuclear domains or nuclear membraneless organelles involved in splicing, thereby increasing splicing efficiency[60].

During the last years it is becoming clear that, in addition to alternative splicing, other regulatory processes related to splicing play an important role in gene expression regulation. One clear example is retention of specific introns that causes nonsense-mediated mRNA decay and nuclear sequestration (recently reviewed in ref. [60]). We anticipate that control of gene expression through modulation of whole gene splicing rate is going to be an important mechanism that will be very much explored in the near future. In addition, our data suggest that pre-mRNAs for many transcription factors encoding genes present a very inefficient splicing. It is possible that some of these relatively stable pre-mRNAs play regulatory roles acting as regulatory long coding RNAs (rlcRNA).

## Methods

### ChrRNA-seq and RNA-seq data processing.
Chromatin-associated RNA-seq (ChrRNA-seq) and total RNA-seq data from NMuMG cells after 2 h or 12 h of TGFβ treatment, or vehicle as control, were obtained from GSE140552[31]. Ribosomal RNA from Chromatin-associated or total RNA samples were depleted using Ribominus technology (Thermo Fisher) and libraries were prepared with the TruSeq Stranded TOTAL RNA kit (Illumina). To process data, reads were aligned to the mm9 mouse reference genome using subjunc function from Rsubread package[61]; TH1 = 2 and unique = TRUE parameters were used. The downstream

analysis was performed on bamfiles with duplicates removed using the samtools[62] rmdup command. IGV tools 2.8.2[63] was used to visualize chromatin-associated and total RNA-seq data.

### Transcriptome annotation file building.
To better fit the transcriptome information to our data, a new transcriptome annotation file was built using our total RNA-seq data (all conditions and replicates merged) and the guide of the transcriptome annotation file from ENSEMBL (including only protein coding genes). To achieve that, Stringtie[64] with --rf and default parameters and GffCompare[65] with -T, -K and default parameters were used. After that, transcripts were filtered including only transcripts with class_code {=, c, j}. Resulting GTF file was used for all downstream analysis. To avoid considering multiple transcripts for each gene, a new GTF considering only the longest transcript was used for $ISI_j$ and $ISI_c$ computation.

### Junction-based ISI calculation.
To compute junction-based ISI ($ISI_j$) for individual introns of expressed genes in vehicle condition squid.py (https://github.com/Xinglab/SQUID) was used with default parameters and considering each replicate separately. The output PI_junction value, for each intron from squid.py corresponds to the proportion of intron inclusion, therefore to calculate intron splicing efficiency $ISI_j$ we subtracted it from 1:

$$ISI_j = 1 - \text{PI\_junction} = 1 - \frac{(EI + IE) \times 0.5}{(EI + IE) \times 0.5 + EE} \tag{1}$$

where $EI$ and $IE$ are the number of exon–intron junction reads and $EE$ is the number of exon–exon junction reads.

To study the relationship of $ISI_j$ value and distance to polyA sites all introns were divided into deciles according to polyA distance and $ISI_j$ distributions were plotted for each decile. To assess gene $ISI_j$ variance from random introns, $ISI_j$ values for individual introns were randomized using sample() function from R, mantaining the number of introns per gene.

### Coverage-based ISI calculation.
To compute coverage-based ISI ($ISI_c$) for individual introns, first a new GTF file with intron coordinates information was created from our previous GTF. Then, ChrRNA-seq reads from vehicle condition (after merging replicates) were assigned to exon or intron independently using feature-Counts function from Rsubread package with following parameters: GTF.featureType = "exon", GTF.attrType = "exon" and strandSpecific = 2, or GTF.featureType = "intron", GTF.attrType = "intron" and strandSpecific = 2, respectively. Then, RPKM (reads per kilobase per million mapped reads) values were obtained using exon or intron length and total mapped reads information. After that, $ISI_c$ was computed as follow:

$$ISI_c = 1 - \frac{I_i}{(E_i + E_{i+1}) \times 0.5} \tag{2}$$

where $I_i$ is the RPKM value for intron $i$ and $E_i$ and $E_{i+1}$ are RPKM values for adjacent exons. To study the relationship of $ISI_c$ value and distance to polyA sites all introns were divided into deciles according to polyA distance and $ISI_c$ distributions were plotted for each decile. To assess gene $ISI_c$ variance from random introns, $ISI_c$ values for individual introns were randomized using sample() function from R, maintaining the number of introns per gene. Only expressed genes were considered. For Supplementary Fig. 2 only genes of the indicated number of introns were used. For randomization introns from genes of the indicated number of introns were used.

### Gene Splicing Index calculation.
To compute Gene Splicing Index (GSI) ChrRNA-seq reads were assigned to exons (considering exons coordinates) or pre-mRNA (considering whole transcripts coordinates) using featureCounts function from Rsubread package with following parameters: GTF.featureType = "exon", GTF.attrType = "gene_id" and strandSpecific = 2, or GTF.featureType = "transcript", GTF.attrType = "gene_id" and strandSpecific = 2, respectively. Then, GSI was computed as follow:

$$GSI_{\text{gene A}} = \log_2 \frac{\frac{\sum_{i=1}^{n} \text{reads exon}_i}{\sum_{i=1}^{n} \text{length exon}_i}}{\frac{\text{reads pre-mRNA}}{\text{length pre-mRNA}}} \tag{3}$$

with $i = 1, 2, …, n$ (number of exons of gene A)

For Figs. 2 and 3 GSI was calculated independently for vehicle conditions after merging replicates. To study the effect of GSI, all expressed genes were divided into deciles according to its GSI value. All comparisons were done using genes from first decile (low-GSI genes) against last decile (high-GSI genes). Only expressing genes were considered.

Differential GSI (ΔGSI) from TGFβ2h or TGFβ12h versus vehicle (data from[31]) was calculated as follow: we performed a differential analysis using linear GSI values ($2^{GSI}$) from two independent replicates of each condition, using a typical voom/limma pipeline[42]. To select those genes with a differential GSI we choose as cutoff | ΔGSI | ≥ 0.5 and p-value < 0.05.

**Gene Ontology**. Gene Ontology analysis (GO) for biological process was performed using DAVID tools (https://david.ncifcrf.gov/).

**Differential gene expression analysis**. To assess differential gene expression, first FeatureCounts() function from Rsubread package was used to assign reads to our newly created GTF annotation file using GTF.featureType = "exon", GTF.attrType = "gene_id" and strandSpecific = 2 parameters on duplicate removed bamfiles. Then differential gene expression analysis was performed using the voom/limma (v.3.34.9) and edgeR (v.3.20.9) Bioconductor packages[66]. CalcNormFactors() function using TMM method was used to normalize samples.

**L x E matrix construction**. To create matrix according to length and pre-mRNA levels, genes were ordered according increasing length or pre-mRNA levels and divided into deciles for length ($L_i$, with $i = 1, 2, …, 10$) and pre-mRNA levels ($E_j$, with $j = 1, 2, …, 10$), separately. Then matrix ($L \times E$) was constructed by assigning genes to the corresponding positions $a_{i,j}$ according to their respective length ($L_i$) and pre-mRNA level ($E_j$). Mean GSI, mean 5' and 3' ss strength and differential GC content of the genes corresponding to each position of the matrix were computed and depicted in a heatmap.

**Gradient vector analysis**. Given the $L \times E$ matrix defined above, where elements $a_{i,j}$ are values of a scalar variable, a vector field is an assignment of a vector to each position $a_{i,j}$ of the matrix. A gradient vector is a vector field that indicates for each position $a_{i,j}$ (with $i$ or $j \neq 1$; and $i$ or $j \neq 10$) the direction in which the matrix $L \times E$ varies more quickly and its module (vector length) represents the rate of variation in that direction. Computation of gradient vector field was performed using raster(), CRS(), persp() and vectorplot() from raster, rasterVis, and RnetCDF R packages.

**Motif strength computation**. Maximum Entropy Model was used to compute motif strength for 5' and 3' splice sites according to ref. [67]. Nine nucleotides sequence (3 bases in exon and 6 bases in intron) and 23 bases (20 bases in intron and 3 bases in exon) were used to compute 5' and 3' ss score, respectively. First and last intron for each transcript were excluded from this analysis.

**Differential GC content**. Bedtools nuc[68] was used to compute GC content for exons and introns separately. Differential GC content for each gene was calculated as: %GC_exons/%GC_introns.

**Model construction**. lm() function from R was used to perform multiple linear regression considering two or more input parameters. Models were constructed including increasing number of input parameters for all considered genes, short genes (genes included in the two first deciles of genes ordered according to increasing size) and long genes (genes included in the two last deciles of genes ordered according to increasing size). Percentage of the variance of the output parameter (GSI) was computed using calc.relimp() function from realimpo R package.

**Cell culture and treatments**. Normal murine mammary gland NMuMG cells (provided by José Antonio Pintor-Toro, CABIMER, in 2014) were cultured in Dulbecco's modified Eagle's medium containing 10% fetal bovine serum (FBS) and 10 µg/ml insulin (complete medium). Cells were tested for mycoplasma contamination periodically. For TGFβ treatments, 5 ng/ml TGFβ1 diluted in 4 mM HCl, 1 mg/ml BSA (240-B, R&D Systems) or 4 mM HCl 1 mg/ml BSA (vehicle, as control), was added to the medium for the indicated time. All TGFβ treatments were performed after 6 h of serum starvation.

**Cellular fractionation, RNA extraction, and RT-qPCR**. Cellular fractionation was carried out according to ref. [69], based in the protocol described in ref. [12]. After treatments, cells were trypsinized and cell pellets were resuspended in 400 µl cold cytoplasmic lysis buffer (0.15% NP-40, 10 mM Tris pH 7.5, 150 mM NaCl) and incubated on ice for 5 min. The lysates were layered onto 1 ml cold sucrose buffer (10 mM Tris pH 7.5, 150 mM NaCl, 24% sucrose w/v), and centrifuged in microfuge tubes at $3500 \times g$ for 10 min. The supernatant containing cytoplasmic fraction was centrifuged at $14,000 \times g$ and stored at 4 °C until obtention of chromatin fraction. The nuclear pellets were gently resuspended into 250 µl cold glycerol buffer (20 mM Tris pH 7.9, 75 mM NaCl, 0.5 mM EDTA, 50% glycerol). An additional 250 µl of cold nuclei lysis buffer (20 mM HEPES pH 7.6, 7.5 mM MgCl₂, 0.2 mM EDTA, 0.3 M NaCl, 1 M urea, 1% NP-40, 1 mM DTT) was added to the samples, followed by a pulsed vortexing and incubation on ice for 2 min. Samples were then spun in microfuge tubes for 2 min at $13,000 \times g$. Fifty microliters of cold phosphate-buffered saline (PBS) was added to the remaining chromatin pellet, and gently pipetted up and down over the pellet, followed by a brief vortex. Then RNA was extracted from cytoplasmic and chromatin fractions using TRIZOL and treated with RQI RNAse-free DNAse (Promega) for DNA removal according to manufacturer's instructions.

Complementary DNA (cDNA) was generated from 2 µg of total RNA using MultiScribe Reverse Retrotranscriptase (Thermo Fisher) following manufacturers instructions. Then, 2 µl of generated cDNA solution was used as a template for real-time PCR (qPCR). Gene products were quantified by qPCR with the Applied Biosystems 7500 FAST Real-Time PCR System, using Applied Biosystems Power SYBR Green Master Mix. Values were normalized to the expression of the *Chd8* gene. At least four biological independent replicates and two technical determinations were performed in each case. All oligonucleotide sequences used are listed in Supplementary Table 1.

**Statistics and reproducibility**. Statistical and graphical data analyses were performed using either Prism 8 (Graphpad) software or R package. To determine the significance between two groups, comparisons were made using two-tailed unpaired Mann–Whitney non-parametric test for $n < 30$. For $n \geq 30$ the Central Limit Theorem indicates that the distribution is approximately Gaussian and then, a two-tailed, paired or unpaired Student *t*-test was used. Number of biological independent replicates is indicated in the figure legends. For all RT-qPCR experiments, four biological independent replicates and two technical determinations of each were performed. For statistical test and standard error determination, only independent replicates were considered. For correlation, Pearson coefficient was calculated using Prism 8 (Graphpad). The horizontal black line of the boxplot represents the median value, the box spans the 25th and 75th percentiles, and whiskers indicate 5th and 95th percentiles. Sample size ($n$) used to derive statistics of all set of data are provided in Supplementary Data 4.

**Reporting summary**. Further information on research design is available in the Nature Research Reporting Summary linked to this article.

## Data availability

The datasets supporting the conclusions of this article are available in Gene Expression Omnibus database (GEO) with the accession number GSE140552 and have been published in ref. [31]. Source data for all graphs and charts are provided in Supplementary Data 5.

## Code availability

We used the following software in the computational analysis: R v3.4.4 (https://cran.r-project.org/bin/linux/ubuntu/), RStudio v0.99.879, (https://rstudio.com/products/rstudio/download/), RSubread v1.28.1 (https://bioconductor.org/packages/release/bioc/html/Rsubread.html), Bioconductor v2.38 (https://www.bioconductor.org/install/) Stringtie. (https://ccb.jhu.edu/software/stringtie/), GffCompare. (https://ccb.jhu.edu/software/stringtie/gffcompare.shtml), squid.py (https://github.com/Xinglab/SQUID), DAVID tools (https://david.ncifcrf.gov/)., Limma-Voom (v.3.34.9) (https://ucdavis-bioinformatics-training.github.io/2018-June-RNA-Seq-Workshop/thursday/DE.html), edgeR (v.3.20.9) (https://bioconductor.org/packages/release/bioc/html/edgeR.html, https://www.rdocumentation.org/packages/edgeR/versions/3.14.0/topics/calcNormFactors), rasterVis (https://cran.r-project.org/web/packages/rasterVis/rasterVis.pdf), RnetCDF (https://www.unidata.ucar.edu/software/netcdf/) and bedtools v2.27.1 (https://bedtools.readthedocs.io/en/latest/content/installation.html).

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

## Acknowledgements

We thank Daniel Rico and Carles Suñé for critical reading of the manuscript, and E. Andújar and M. Pérez from the CABIMER Genomic Unit for technical assistance. This work was funded by the Spanish Ministry of Economy and Competitiveness MCIN/AEI/ 10.13039/501100011033/ (grant numbers BFU2017-85420-R and PID2020-118516GB-I00), the Junta de Andalucía (grant P18-FR-1962 and BIO-321), "Fundación Vencer El Cancer" (VEC) and the European Union FEDER "A way to build Europe". CABIMER is a Center partially funded by the Junta de Andalucía.

## Author contributions

Conceptualization and methodology, J.A.G.-M. and J.C.R.; Investigation, E.S.-E. and J.A.G.-M.; Writing, J.C.R. and J.A.G.-M.; Funding acquisition: J.C.R.

## Competing interests

The authors declare no competing interests.
