## [Peer Review File · Communications Biology]

Reviewers' comments:

Reviewer #1 (Remarks to the Author):

In the article by Elena et al., the authors assessed co-transcriptional splicing (CTS) efficiency of different genes with nascent chromatin-associated RNA sequencing data. CTS is an important and complicated process. The article reported associations between CTS and gene length, RNA abundance, and gene functions. They also showed that TGFbeta treatment results in CTS changes of specific genes. In general, the study is descriptive, without new insights into specific machineries. Following are some major questions and concerns I have.

1. From Fig. 1D, it appears that the ISlc values depend on numbers of introns rather than the host genes. In other words, genes with similar numbers of introns tend to have similar ISlc values. Therefore, randomly selected intron sets are not proper controls for the comparisons in Fig. 1E, F.
2. RPKM values could be highly biased for short RNA fragments. Therefore, for short introns and exons, the authors should be very careful when using RPKM to calculate the levels of GSI and ISlc. They should either filter out the short introns and exons or figure out a way to address the RPKM bias due to length differences.
3. What does Fig. 2E mean in biology? These GO terms are very general. It is unclear how GSI values are associated to these very general functions. Important information is missing, for example, how many genes in a specific term were actually GSI high or low? One has to suspect that these apparent enrichments are just artifacts of the GO analyses.
4. From Fig. 3A, I am not convinced that GSI is negatively correlated with pre-mRNA levels.
5. Fig. 3C: I am suspecting that the correlation between GSI and gene length is potentially due to the RPKM bias as mentioned in comment #2. If the authors truncate the lengthy genes to generate "fake" genes, would they get the same GSI values as the full length genes? This needs to be carefully addresses, as the authors claims that gene length is the strongest factor defining the GSI values.
6. A major problem is that apparently, no biological replicate was included in the TGFbeta study. The authors simply used deltaGSI of 0.5 as cutoffs. How can we tell whether or not such a small change represents a biologically relevant machinery rather than just technical noise? After all, GSI ranges from 0 to 6. More replicates are definitely needed to differentiate, to some extent, the real biological changes and technical variations.

Reviewer #2 (Remarks to the Author):

In this manuscript, authors utilized nascent chromatin-associated RNA-sequencing data recently published (Guerrero-Martinez et al., 2020, Nature communications 11: 6196) to study the efficiency of co-transcriptional splicing. They found that co-transcriptional splicing tends to be similar between the different introns of a gene and that genes that are short and abundant in terms of pre-mRNA expression undergo inefficient splicing, while genes that are long and less abundant in terms of pre-mRNA exhibit a more efficient splicing. Furthermore, authors revealed that the TGFb pathway regulates the co-transcriptional splicing efficiency by causing changes in the levels of mRNA. The topic of the paper is interesting, and the manuscript is well written.

Major

- What about the role of the size of the exons and the size of the introns? Would this affect the results and interpretations?
- The following statement from the authors is a good summary of the entire paper: "two extreme behaviors: highly transcribed short genes encoding mostly transcription factors or translation proteins, which display inefficient splicing, and long genes mostly encoding metabolic enzymes or transport proteins with a not very high level of pre-mRNA" (lines 260-263). Could you please discuss better which the biological and physiological implications of this are?
- As it is the paper is mainly bioinformatics, what is OK. However, I think it can be significantly improved if authors discuss or even show some evidence of the implications of their findings in the biology of the cells or within a physiological context. In that manner, the readers and the scientific community will appreciate even more the importance of their discoveries.

First we want to thank Reviewers for their interesting and appropriated comments and suggestions. We think we have addressed most of the Reviewer's questions and the revised manuscript has improved very much in quality. Answers to the Reviewer's comments are in boldface. Figures for the Reviewers, designed Figures R1, R2...etc.

Reviewer #1 (Remarks to the Author):

In the article by Elena et al., the authors assessed co-transcriptional splicing (CTS) efficiency of different genes with nascent chromatin-associated RNA sequencing data. CTS is an important and complicated process. The article reported associations between CTS and gene length, RNA abundance, and gene functions. They also showed that TGFbeta treatment results in CTS changes of specific genes. In general, the study is descriptive, without new insights into specific machineries. Following are some major questions and concerns I have.

1. From Fig. 1D, it appears that the ISlc values depend on numbers of introns rather than the host genes. In other words, genes with similar numbers of introns tend to have similar ISlc values. Therefore, randomly selected intron sets are not proper controls for the comparisons in Fig. 1E, F.

Since the number of introns is a characteristic of each gene, we understand that it is correct to state that "This result suggested that CTS efficiency is, at least in part, a gene-specific trait". Nevertheless, we understand the concern of the Reviewer and we have repeated the calculation performed for Figure 1E using 7 subsets of genes with similar number of introns (2-5, 6-10, 11-15, 16-20, 21-25, 26-35, >35) (Figure R1). Obviously, in this case, intron randomization was performed only among introns belonging to genes with a similar number of introns. As can be observed in the figure,

this analysis again demonstrated that ISI levels of introns within the same gene tended to be more similar to each other than those of introns from different genes. We have included this analysis in the manuscript as Supplementary Figure 2 (text in lines 168-170, lines 545-547 and lines 986-989).

Figure R1. Variance of ISlc values across introns within the same gene or the same number of randomly sampled introns. In each plot only genes with the indicated number of introns were used. Unpaired Student's

t-test p-values are shown.

2. RPKM values could be highly biased for short RNA fragments. Therefore, for short introns and exons, the authors should be very careful when using RPKM to calculate the levels of GSI and ISlc. They should either filter out the short introns and exons or figure out a way to address the RPKM bias due to length differences.

First we have carefully reviewed the bibliography about length-dependent RPKM bias.

In fact, several papers analyze the effect of length on RPKM when performing comparison between samples. Thus, the total number of reads for a given transcript is proportional to the expression level of the transcript but also to the length of the transcript. In other words, a long transcript will have more reads mapped to it compared to a short gene of similar expression. Therefore, since the statistical power of an experiment is proportional to the sampling size, there is more power to detect differential expression for longer genes (Mandelbom *et al*, 2019; Oshlack & Wakefield, 2009; Zhao *et al*, 2020). Small differences between samples during the preparation of the libraries (specially degree of fragmentation) or total number of mapped reads may affect in slightly different way to long and short genes when doing differential expression analysis. However, in the first part of our manuscript (figure 1 to 3) we do not compare between samples but between genes within the same sample. In the second part (Figures 4 to 6) we compare between samples (vehicle vs TGFb2h or vehicle vs TGFb12h), but in this case gene length is not a variable in our analysis.

Nevertheless, we have investigated in detail the existence of the bias indicated by the Reviewer in our data:

To investigate how the exon or intron length affects RPKM values we have binned the exons and introns into deciles depending on their length. Then, we have plotted the RPKM of exons (Figure R2A) or Introns (Figure R2B) of each one of the different deciles.

Figure R2. A. Boxplots showing exon RPKM distribution of exon with the indicated length. Exons were divided into ten deciles depending on their length. B Boxplots showing intron RPKM distribution of introns with the indicated intron length. Introns were divided into ten deciles depending on their length.

Figure R3. Boxplots showing distribution of lengths of genes containing introns with the indicated size.

As observed in Figure R2A there is not a specific bias for short exons. In fact, long exons (deciles 9 and 10) have similar median than short exons (decile 1). This plot is perfectly consistent with the current knowledge about the effect of exon size in alternative splicing and exon skipping. Thus, too short exons present steric hindrance that may interfere with the requirements of the splicing machinery. For example, *Sterner et al.* reported that a constitutively recognized internal exon is frequently skipped *in vivo* by the splicing machinery if its size is reduced to less than 50 nucleotides (Dominski & Kole, 1991). As a consequence, very short exons are often skipped (Zheng *et al.*, 2005) leading to decreased RPKM. On the other hand, it is well known since the nineties that spliceosome formation is also strongly inhibited if exons lengths are expanded artificially to over 300 nucleotides (Robberson *et al.*, 1990; Sterner *et al.*, 1996). Therefore, and interestingly, despite this plot (Figure R2A) is not representing splicing efficiency, but accumulated RPKM in exons of the indicated size, the fact that very short and very long exons are poorly spliced probably causes reduced inclusion and therefore, they are under-represented in the transcriptome. Consequently, we conclude that we have not a bias of RPKM in small exons.

Short introns have a tendency to have higher values of RPKM (Figure R2B). Since short exons present lower RPKM than medium size exons, and short introns display higher RPKM than medium size introns, the bias should not be a consequence of the RPKM calculation, but maybe an intrinsic characteristic of short introns. This fact may be due to a combination of the following three factors: 1) short introns have a slower splicing kinetics (Wachutka *et al.*, 2019). 2) Genes with short introns are more transcribed. In fact, it has been shown that there is a selection for short introns in highly expressed genes (Castillo-Davis *et al.*, 2002). 3) Genes with short introns present worse splicing efficiency. We provide evidences in the manuscript that short genes tend to have worse splicing, since short introns tend to be in short genes (Figure R3), our results support possibility number 3, which may be also related to 1 and 2. In any case, high RPKM in short introns does not seem to be an artefact of the RPKM calculation. Therefore, we have not been able to find a positive or negative bias in RPKM in short exons or introns that cannot be explained by the intrinsic properties of short introns and exons.

Finally, concerning the GSI calculation. Thanks to Reviewer #1 comment we revised all our RPKM and GSI calculations. We have realized that the formula we wrote in the original manuscript was not the one we actually used to calculate GSI. The correct formula is:

$$GSI_{gene A} = \log_2 \frac{\frac{\sum_{i=1}^n reads\ exon_i}{\sum_{i=1}^n length\ exon_i}}{\frac{reads\ pre_mRNA}{length\ pre_mRNA}}$$

with $i = 1, 2, \dots, n$ (number of exons of gene A)

Therefore, for numerator we do not calculate RPKM of each exon but we add reads of all exons of a gene and divide by the sum of lengths of all exons of the gene (the length of the mature transcript). Therefore, there is not RPKM of small introns or small exons in the computation of GSI. We are sorry for the mistake.

While this is a very interesting discussion we think it is out of the scope of our manuscript and we have not included this analysis in the revised manuscript text. Of course, we have corrected the mistake in the formula for GSI calculation (MS text modified in lines 558-560).

3. What does Fig. 2E mean in biology? These GO terms are very general. It is unclear how GSI values are associated to these very general functions. Important information is missing, for example, how many genes in a specific term were actually GSI high or low? One has to suspect that these apparent enrichments are just artifacts of the GO analyses.

The number of genes in each category has been now included in the figure 2E, as indicated by the reviewer. We have used the GOTERM_BP_ALL option of DAVIS database which includes both general and specific GO terms. We have selected general terms because many more genes are included, which increases significance. As can be observed in the new figure, many High-GSI or Low-GSI genes belong to the indicated categories. In addition, p-values demonstrate that enrichments are very significant (about 10^{-5} to 10^{-22}), because of that we do not think that this is an artifact.

Which could be the biological meaning of these enrichments? We think this is mostly due to differences in length of the genes from the different categories. Thus, ribosomal proteins and many transcription factors are encoded by short genes (Heyn *et al*, 2015; Tullai *et al*, 2007; Yoshihama *et al*, 2002). Since low-GSI genes tend to be small genes, this fact may explain that translation and transcription GO categories are enriched among the low-GSI genes. In contrast, extracellular matrix and adhesion molecules, transport and endocytosis proteins, and many enzymes are often encoded by long genes. Since high-GSI genes tend to be long genes, this fact would explain the enrichment of these categories among the high-GSI genes.

Gene length bias in specific functional sets of genes has been already described. For example, it has been shown that mouse and human brains express a greater proportion of long genes relative to non-neural tissues (Gabel *et al*, 2015; Zylka *et al*, 2015). Thus, central nervous system expresses many cell adhesion, axon guidance and synapse formation proteins that are encoded by long genes. Furthermore, it has been shown that the size of the genes among autism candidate genes is biased towards large genes (Shohat & Shifman, 2014).

Confirming this argument, Figure R4 shows that, the genes included in the “cell adhesion”, “transport” and “endocytosis” GO terms, in average present larger length than the genes included in “translation”, “transcription” and “regulation of transcriptions” GO terms. We think that this bias in gene length of specific GO categories may explain the functional enrichments observed. We have better clarified in the text of the revised manuscript the possible reasons for these GO enrichments and

Figure R4 has been included as Supplementary Figure 7. (MS text modified in lines 413-419 and line 900 and lines 1020-1021)

Figure R4. Boxplots showing gene length distributions of genes of the indicated GO terms. Number of genes in each term is indicated.

4. From Fig. 3A, I am not convinced that GSI is negatively correlated with pre-mRNA levels.

We have calculated the Pearson correlation coefficient of the data presented in figure 3A: $R = -0.171$ and $p < 0.0001$ using GraphPad Prism, indicating that there is a negative correlation. We have included these data in the manuscript (line 196). Nevertheless, we understand that for Pearson Correlation, the P-value is the probability that you would have found the current result if the correlation coefficient were in fact zero (null hypothesis). Given the large amount of data, p-value is going to be very low even if the correlation is not very good. Because of that, we accompanied the scatter plot of figure 3A with a boxplot (Figure 3B) showing the very important expression difference between the low-GSI and the high-GSI genes. In this case the difference is very significant with a p-value = 2.8×10^{-31} .

5. Fig. 3C: I am suspecting that the correlation between GSI and gene length is

potentially due to the RPKM bias as mentioned in comment #2.

In the answer to comment #2, we have shown that there is not a specific bias that significantly increases or decreases the values of RPKM when quantified in small exons or introns. We also discussed that, in the bibliography, RPKM length bias is only an issue when considering differential expression of a gene between samples, not for different genes in the same sample. Finally, we have also indicated that computation formula of GSI does not use RPKM of small exons or introns.

If the authors truncate the lengthy genes to generate “fake” genes, would they get the same GSI values as the full length genes? This needs to be carefully addresses, as the authors claims that gene length is the strongest factor defining the GSI values.

We have carefully performed the analysis that Reviewer #1 suggests. Figure R5 shows the effect of dividing into two halves of identical size each gene. As can be observed, the positive correlation between GSI and gene length is maintained; however, the slope of the regression line, and the correlation coefficient decreases. This decrease of correlation and slope is mainly consequence of two facts: 1) the lack of homogenous distribution of exons and introns which can produces differences of the two new calculated GSI and 2) the new calculated GSI values are close but not identical to the GSI of the real gene due to small differences in the signal of each part. Both factors necessarily decrease correlation coefficient and slope because increase dispersion of the data.

Figure R5. Correlation between gene length and GSI values. A. Figure 3c of the manuscript using “real” genes. B. Same that A but genes are divided into two halves with identical size.

GSI was calculated using reads and length of the two new “fake” genes and length of the new “fake” genes was consigned in the y axis.

Figure R6. Examples of calculation of GSI of truncated fake genes.

Figure R6 shows some examples to better understand the different cases. Let's call GSI₁ and GSI₂ to the GSI values of the newly generated 5' and 3' gene halves. For example, *Fos* is a small gene with only four exons and three introns that has a low GSI (0.31). Division into two creates two halves with very different intron-exon composition and with GSI₁=0.64 and GSI₂=0.08. The average of both values is close to the GSI of the real gene, but intrinsic variability of the RNA-seq signal increases dispersion in the correlation, respect to the original value. For a long gene with many exons and introns, like *Utrm*, the values change from GSI = 3.72 to GSI₁ = 3.45 and GSI₂ = 3.69, if the gene is divided into two halves and GSI₁ = 3.49, GSI₂ = 3.90

and $GSI_3 = 3.69$, if the gene is divided into three equal parts. Again, *Utrm* is a gene with high GSI, and the new values remains high, but there are small changes that introduce dispersion. For *Nap1l4*, $GSI = 1.11$ for the complete gene. The average of the partial GSIs after dividing into 2 or into three parts is close to the full length GSI ($(1.60+0.85)/2 = 1.22$ and $(1.35+2.42+0.25)/3 = 1.34$), but again the dispersion of new calculated data will decrease the correlation coefficient and the slope. An extreme case is the gene *Dagla*. It is quite common, specially in mammals, that genes have very long first introns (Bradnam & Korf, 2008; Park *et al*, 2014). It often occurs that the first intron expands more than half of the gene. In the case of *Dagla*, truncation of the gene into two halves causes that the 5' half has only the first exon (58 nucleotides) and an about 30 kb intron (99,8% of the sequence). Very often signal of the annotated first exons is not well represented in the RNA-seq (Figure R6 for the *Dagla* gene). As a consequence, the GSI of the first half is negative ($GSI_1 = -1.19$) which is clearly an artefact.

Figure R6. ChrRNA-seq IGV snapshot of *Dagla* gene exon 1.

Most of the times the average of the “fake” GSIs is similar the real GSI (Figure R7). Figure R8 shows that even if average the fake GSIs is exactly equal to the real GSI, the two new data will decrease slope of the regression curve. Slope decrease will be larger depending on the dispersion (standard deviation) of the two “fake” data.

Figure R7. Distribution of $GSI-(GSI_1+GSI_2)/2$ values. This demonstrate that, in most of the cases, the average of the new GSIs is very similar to the real GSI.

Figure R8. Simulation of the effect of data dispersion on slope of regression line. (a, f) Original plot. (b, g) $Y'=Y/2$ and identical X ($X=X'$). (c, h) $Y'=Y/2$ and $(X'1+X'2)/2=X$, with small dispersion. (d, i) $Y'=Y/2$ and $(X'1+X'2)/2=X$, with medium dispersion. (e, j) $Y'=Y/2$ and $(X'1+X'2)/2=X$ with large dispersion. (a-

e) Linear variable. (f-j) Logarithmic variable.

In summary, the GSI of the truncated “fake” genes is often close but not identical to the real GSI value. Most of the times the average of the “fake” GSIs is similar the real GSI. But sometimes artefacts occur after the truncation. All these possibilities cause dispersion of the data and a decrease in the correlation and the slope of the regression curve, but the positive correlation between gene length and GSI is maintained. Since sometimes truncation increases GSI and in other cases decreases GSI, we do not see any constant bias due to length in the calculation of GSI.

6. A major problem is that apparently, no biological replicate was included in the TGFbeta study. The authors simply used deltaGSI of 0.5 as cutoffs. How can we tell whether or not such a small change represents a biologically relevant machinery rather than just technical noise? After all, GSI ranges from 0 to 6. More replicates are definitely needed to differentiate, to some extent, the real biological changes and technical variations.

Reviewer #1 is right. In fact, we have used the data of two independent chromatin-RNA-seq experiments (Guerrero-Martinez *et al*, 2020), but the analysis performed in the original version of the manuscript was carried out with the addition of the reads of both experiments together. Now we have performed the analysis of each chromatin-RNA-seq experiment separately and applied differential expression statistics (LIMMA) using the GSI data. The cutoff was established in $|\Delta\text{GSI}| \geq 0.5$ with a p-value < 0.05 . Only 325 of the original 544 genes passed the new cutoff. Therefore, all panels of Figures 4 and 5 were remade with the new data. The conclusions are identical than in the original manuscript. All model genes that were selected for Figure 4E, Supplementary Figure 6a, 6b (revised version numbering) and Figures 6A and 6E passed the new cutoff. (MS text modified in lines 284 to 289 and 567-572). Now the data are much more robust and we thank the reviewer for this comment.

Reviewer #2

In this manuscript, authors utilized nascent chromatin-associated RNA-sequencing data recently published (Guerrero-Martinez *et al.*, 2020, Nature communications 11: 6196) to study the efficiency of co-transcriptional splicing. They found that co-transcriptional splicing tends to be similar between the different introns of a gene and that genes that are short and abundant in terms of pre-mRNA expression undergo inefficient splicing, while genes that are long and less abundant in terms of pre-mRNA exhibit a more efficient splicing. Furthermore, authors revealed that the TGFb pathway regulates the co-transcriptional splicing efficiency by causing changes in the levels of mRNA. The topic of the paper is interesting, and the manuscript is well written.

Major

- What about the role of the size of the exons and the size of the introns? Would

this affect the results and interpretations?

This is a very interesting question that has been very well studied by others and we have decided not to get into it in the manuscript. We have centered the manuscript in the analysis of GSI, that study the splicing at the whole gene level.

We summarize here what is known about the role of exons and introns size in splicing efficiency. It is well known that very small introns are difficult to splice (Wachutka *et al.*, 2019). Please see answer to comment #2 of Reviewer #1 about this question. It has been observed a general positive correlation of splicing efficiency with introns length (Ameur *et al.*, 2011; Khodor *et al.*, 2011; Pai *et al.*, 2017; Windhager *et al.*, 2012). However, this effect is sometimes masked by the fact that first introns are spliced slowly (see supplementary figure 1 of the manuscript and (Bedi *et al.*, 2021; Pai *et al.*, 2017). Since first introns use to be very long (Tilgner *et al.*, 2012), correlation between splicing efficiency and intron length is better observed if first introns are removed.

In the case of exons it is also clear that very short and longer than 300 bp exons are poorly spliced (Dominski & Kole, 1991; Khodor *et al.*, 2012; Khodor *et al.*, 2011; Li *et al.*, 2020; Robberson *et al.*, 1990; Sterner *et al.*, 1996; Zheng *et al.*, 2005).

Since we have similar results, we think it is unnecessary to extend the manuscript with more data about this already-studied topic. Specially because some of the cited manuscript uses methodologies similar to the one we use here. However, if the reviewer still thinks that our analysis of intron and exon length effect on splicing efficiency should be included in the manuscript we will do it.

• The following statement from the authors is a good summary of the entire paper: “two extreme behaviors: highly transcribed short genes encoding mostly transcription factors or translation proteins, which display inefficient splicing, and long genes mostly encoding metabolic enzymes or transport proteins with a not very high level of pre-mRNA” (lines 260-263). Could you please discuss better which the biological and physiological implications of this are?

Please see answer to point #3 of Reviewer #1. Since high-GSI tend to be long genes and low-GSI tend to be small genes, we think that the Gene Ontology categories enriched among the high-GSI and low-GSI genes are related to the length of the genes of these categories. Ribosomal proteins and many transcription factors, specially, immediate early transcription factors are encoded by short genes with a reduced number of introns (see figure R4) (Tullai *et al.*, 2007; Yoshihama *et al.*, 2002). In contrast, transport, endocytosis and cell adhesion categories are constituted by, in average, longer genes (see figure R4). A commentary about this has been now included in the Discussion section of the revised version. (MS text modified in lines 413-419 and line 900 and lines 1020-1021)

• As it is the paper is mainly bioinformatics, what is OK. However, I think it can be significantly improved if authors discuss or even show some evidence of the implications of their findings in the biology of the cells or within a physiological context. In that manner, the readers and the scientific community will appreciate even more the importance of their discoveries.

We have shown that gene splicing efficiency can be modulated by TGFbeta, which it is a physiological signaling pathway. We also show transient changes in splicing efficiency after TGFbeta stimulation which we suggest it may constitute a splicing-dependent temporal regulation of gene expression. In the original version of the manuscript we also discussed the possibility that the fact that long genes present an efficient splicing may have an energetic implication:

“From a metabolic and energetic perspective, synthesis of long-gene pre-mRNAs requires massive quantities of nucleotides, energy and time. It seems reasonable to suppose that long genes have been subjected to a strong selective pressure to have a very efficient splicing process.”

Now, in the revised version, following the suggestion of the reviewer we have added a sentence in the discussion summarizing potential implications:

1) Inefficient splicing is normally associated to constitutive delayed expression but it can be in fact an additional way of regulating gene expression. It is possible that the existence of a pool of pre-mRNA offers additional ways of regulation of gene expression by controlling the splicing rate.

2) The existence of very inefficient splicing in certain pre-mRNAs suggest that these RNAs may have regulatory roles per se, acting as regulatory long coding RNAs (rlcRNA).

(MS text modified in lines 467 to 477)

REFERENCES

Ameur A, Zaghlool A, Halvardson J, Wetterbom A, Gyllensten U, Cavelier L, Feuk L (2011) Total RNA sequencing reveals nascent transcription and widespread co-transcriptional splicing in the human brain. *Nat Struct Mol Biol* 18: 1435-1440

Bedi K, Magnuson BR, Narayanan I, Paulsen M, Wilson TE, Ljungman M (2021) Co-transcriptional splicing efficiencies differ within genes and between cell types. *RNA (New York, NY)*

Bradnam KR, Korf I (2008) Longer first introns are a general property of eukaryotic gene structure. *PLoS ONE* 3: e3093

Castillo-Davis CI, Mekhedov SL, Hartl DL, Koonin EV, Kondrashov FA (2002) Selection for short introns in highly expressed genes. *Nat Genet* 31: 415-418

Dominski Z, Kole R (1991) Selection of splice sites in pre-mRNAs with short internal exons. *Mol Cell Biol* 11: 6075-6083

Gabel HW, Kinde B, Stroud H, Gilbert CS, Harmin DA, Kastan NR, Hemberg M, Ebert DH, Greenberg ME (2015) Disruption of DNA-methylation-dependent long gene repression in Rett syndrome. *Nature* 522: 89-93

Guerrero-Martinez JA, Ceballos-Chavez M, Koehler F, Peiro S, Reyes JC (2020) TGFbeta promotes widespread enhancer chromatin opening and operates on genomic regulatory domains. *Nature communications* 11: 6196

Heyn P, Kalinka AT, Tomancak P, Neugebauer KM (2015) Introns and gene expression: cellular constraints, transcriptional regulation, and evolutionary consequences. *Bioessays* 37: 148-154

Khodor YL, Menet JS, Tolan M, Rosbash M (2012) Cotranscriptional splicing efficiency differs dramatically between *Drosophila* and mouse. *RNA (New York, NY)* 18: 2174-2186

Khodor YL, Rodriguez J, Abruzzi KC, Tang CH, Marr MT, 2nd, Rosbash M (2011) Nascent-seq indicates widespread cotranscriptional pre-mRNA splicing in *Drosophila*. *Genes Dev* 25: 2502-2512

Li S, Wang Y, Zhao Y, Zhao X, Chen X, Gong Z (2020) Global Co-transcriptional Splicing in *Arabidopsis* and the Correlation with Splicing Regulation in Mature RNAs. *Mol Plant* 13: 266-277

Mandelboum S, Manber Z, Elroy-Stein O, Elkon R (2019) Recurrent functional misinterpretation of RNA-seq data caused by sample-specific gene length bias. *PLoS Biol* 17: e3000481

Oshlack A, Wakefield MJ (2009) Transcript length bias in RNA-seq data confounds systems biology. *Biol Direct* 4: 14

Pai AA, Henriques T, McCue K, Burkholder A, Adelman K, Burge CB (2017) The kinetics of pre-mRNA splicing in the *Drosophila* genome and the influence of gene architecture. *eLife* 6

Park SG, Hannenhalli S, Choi SS (2014) Conservation in first introns is positively associated with the number of exons within genes and the presence of regulatory epigenetic signals. *BMC Genomics* 15: 526

Petibon C, Malik Ghulam M, Catala M, Abou Elela S (2021) Regulation of ribosomal protein genes: An ordered anarchy. *Wiley interdisciplinary reviews* 12: e1632

Robberson BL, Cote GJ, Berget SM (1990) Exon definition may facilitate splice site selection in RNAs with multiple exons. *Mol Cell Biol* 10: 84-94

Shohat S, Shifman S (2014) Bias towards large genes in autism. *Nature* 512: E1-2

Stern DA, Carlo T, Berget SM (1996) Architectural limits on split genes. *Proc Natl Acad Sci U S A* 93: 15081-15085

Tilgner H, Knowles DG, Johnson R, Davis CA, Chakraborty S, Djebali S, Curado J, Snyder M, Gingeras TR, Guigo R (2012) Deep sequencing of subcellular RNA fractions shows splicing to be predominantly co-transcriptional in the human genome but inefficient for lncRNAs. *Genome Res* 22: 1616-1625

Tullai JW, Schaffer ME, Mullenbrock S, Sholder G, Kasif S, Cooper GM (2007) Immediate-early and delayed primary response genes are distinct in function and genomic architecture. *J Biol Chem* 282: 23981-23995

Wachutka L, Caizzi L, Gagneur J, Cramer P (2019) Global donor and acceptor splicing site kinetics in human cells. *eLife* 8

Windhager L, Bonfert T, Burger K, Ruzsics Z, Krebs S, Kaufmann S, Malterer G, L'Hernault A, Schilhabel M, Schreiber S *et al* (2012) Ultrashort and progressive 4sU-tagging reveals key characteristics of RNA processing at nucleotide resolution. *Genome Res* 22: 2031-2042

Yoshihama M, Uechi T, Asakawa S, Kawasaki K, Kato S, Higa S, Maeda N, Minoshima S, Tanaka T, Shimizu N *et al* (2002) The human ribosomal protein genes: sequencing and comparative analysis of 73 genes. *Genome Res* 12: 379-390

Zhao S, Ye Z, Stanton R (2020) Misuse of RPKM or TPM normalization when comparing across samples and sequencing protocols. *RNA (New York, NY)* 26: 903-909

Zheng CL, Fu XD, Gribskov M (2005) Characteristics and regulatory elements defining constitutive splicing and different modes of alternative splicing in human and mouse. *RNA (New York, NY)* 11: 1777-1787

Zylka MJ, Simon JM, Philpot BD (2015) Gene length matters in neurons. *Neuron* 86: 353-355

REVIEWERS' COMMENTS:

Reviewer #1 (Remarks to the Author):

The revised manuscript has addressed the majority of my previous comments.